# Study on the pathogenesis of *PLXNB1* gene in olfactory dysfunction of allergic rhinitis

**Xinglong Chen[1,2], Lingqiong Zhao[1,3], Wenlong Luo[1]***

**1** Department of Otolaryngology-Head and Neck Surgery, The Second Affiliated Hospital of Chongqing Medical University, Chongqing, China, **2** West China School of Medicine, Sichuan University, Sichuan University Affiliated Chengdu Second People's Hospital, Department of Otolaryngology-Head and Neck Surgery, Chengdu Second People's Hospital, Chengdu, Sichuan, China, **3** Department of Oncology, Chongqing General Hospital, Chongqing, China

* Luowenlong162@163.com

## Abstract

### Background

Olfactory dysfunction (OD) is a common yet underrecognized symptom of allergic rhinitis (AR). While inflammation contributes to OD, the underlying molecular mechanisms remain unclear.

### Methods

Differential gene expression analysis was conducted using GEO datasets (GSE75011 and GSE43523). OD DEGs were identified and used to construct a protein-protein interaction (PPI) network. Key genes were filtered through LASSO regression and validated through immune infiltration analysis. The biological function of *PLXNB1* was further investigated via in vitro human nasal epithelial cell models and an in vivo mouse model of AR.

### Results

*PLXNB1* expression was significantly downregulated in AR patients compared to healthy controls in both training and validation datasets. Immune infiltration analysis revealed a negative correlation between *PLXNB1* and pro-inflammatory immune cells. Functional assays showed that *PLXNB1* knockdown led to increased expression of inflammatory cytokines (e.g., IL-4, IL-6), likely via activation of the *MAPK/p38* signaling pathway. Conversely, overexpression or Desloratadine treatment restored *PLXNB1* levels and suppressed inflammation.

### Conclusion

*PLXNB1* is a potential hub gene linking immune dysregulation to olfactory dysfunction in AR. Its downregulation exacerbates inflammation and may impair olfactory

**Data availability statement:** The datasets analyzed in this study are publicly available in the Gene Expression Omnibus (GEO) database under accession numbers GSE75011 and GSE43523 (https://www.ncbi.nlm.nih.gov/geo/). All other data supporting the findings of this study are available from the corresponding author upon reasonable request.

**Funding:** The author(s) received no specific funding for this work.

**Competing interests:** The authors declare that they have no conflicts of interest.

function via the *MAPK/p38* axis. These findings offer new insights into AR pathogenesis and identify *PLXNB1* as a promising target for therapeutic intervention.

## Introduction

Allergic Rhinitis (AR) is a non-infectious inflammatory disease mediated by IgE, affecting approximately 10% to 20% of the global population [1,2]. AR is a common chronic inflammatory condition primarily caused by hypersensitivity of the nasal mucosa to various allergens such as pollen, dust mites, and animal dander [3–5]. The main clinical symptoms include frequent sneezing, nasal congestion, rhinorrhea, and nasal itching [6]. Patients may experience a decline or loss of olfactory sensitivity, throat discomfort, and eye symptoms such as itching and congestion [7]. Long-term sufferers may also experience fatigue, insomnia, and headaches [8]. Treatment strategies include avoiding allergen exposure, pharmacotherapy (such as antihistamines and corticosteroid nasal sprays), and allergen immunotherapy to alleviate symptoms and improve quality of life [9].

Olfactory Dysfunction (OD) is a common symptom among patients with AR, manifesting as a reduction or complete loss of olfactory perception [10]. This includes decreased olfactory sensitivity, impaired odor discrimination, and diminished odor quality [11,12]. This issue not only affects patients' perception of food, environment, and safety but can also lead to malnutrition, psychological anxiety, and social obstacles [13]. Traditional research on AR has primarily focused on the immune response and inflammatory processes triggered by allergens [14]. However, recent studies suggest that OD may involve more complex molecular mechanisms, including abnormal gene expression, altered neuronal activity, and structural and functional abnormalities of the nasal mucosa.

Bioinformatics plays a crucial role in elucidating the molecular mechanisms underlying OD [15]. By integrating large-scale gene expression data with bioinformatics analysis tools, it is possible to identify and analyze key genes, pathways, and biological processes related to OD in patients with allergic rhinitis. This not only aids in the comprehensive understanding of the disease's pathogenesis but also provides a theoretical basis for the development of personalized therapeutic and diagnostic approaches in the future.

In this study, relevant datasets (GSE75011 and GSE43523) were collected from the GEO public database for differential expression analysis. After screening for olfaction-related differentially expressed genes, a protein-protein interaction (PPI) network of hub genes was constructed. Lasso regression was then employed to further identify key genes. Subsequent expression analysis and receiver operating characteristic (ROC) curve assessment were performed to evaluate diagnostic efficacy. Additionally, single-sample gene set enrichment analysis (ssGSEA) was used to analyze the relationship between diagnostic genes and immune cells. To further verify the bioinformatic findings, we conducted in vitro and in vivo experiments. These included nasal epithelial cell cultures stimulated with allergens, and AR mouse models to evaluate behavioral symptoms, histological changes, and gene/protein

expression of *PLXNB1* and immune markers. Based on the above, we hypothesize that *PLXNB1* is a key gene involved in the olfactory dysfunction of AR patients, potentially mediating this through immune-related pathways.

## Materials and methods

### Identification of olfactory nerve regulation-related genes

To identify genes associated with the regulation of olfactory nerves, we performed a Gene Set Enrichment Analysis (GSEA) using our dataset. We extracted 18 genes related to olfactory nerves from the database for subsequent analysis.

### Ethics statement

This study utilized publicly available datasets from the GEO database, which did not require ethical approval. In addition, animal experiments were conducted as part of this research. All animal procedures were approved by the All animal procedures were approved by the Institutional Animal Care and Use Committee of the Second Affiliated Hospital of Chongqing Medical University (IACUC-SAHCQMU, Approval No. IACUC-SAHCQMU-2025-0156). All experiments were performed in accordance with relevant guidelines and regulations.

### Data collection and preparation

Gene expression profiles were downloaded from the GEO database (http://www.ncbi.nlm.nih.gov/geo). We selected GSE75011 as the training set, which includes 15 control and 25 AR blood samples. The expression profiles were generated on the GPL16791 platform using the NexteraXT Illumina sequencing platform for library preparation. GSE43523, which includes 7 AR cases and 5 healthy control samples, was used as the validation set, based on the GPL6883 platform (Illumina HumanRef-8 v3.0 expression beadchip). The main characteristics of the datasets are listed in Table 1. All data are publicly available online.

### Differential expression analysis

Differential expression analysis was conducted using the R package "DESeq2" [16] to compare gene expression differences between AR patients and healthy controls. Statistical significance was determined by |log2FoldChange| > 0 and adjusted p-value < 0.05. Significant differentially expressed genes (DEGs) related to OD were identified by intersecting DEGs with olfactory nerve genes using a Venn diagram. Subsequently, upregulated and downregulated olfaction-related DEGs were subjected to GO/KEGG enrichment analysis using the R packages "clusterProfiler" [17] and "org.Hs.eg.db," [18] with p-values < 0.05 considered statistically significant.

### PPI network analysis

The identified key DEGs were used to construct a PPI network. PPI analysis was performed using public databases or bioinformatics tools such as STRING, and the resulting interactions were visualized as networks using Cytoscape (version 3.9.1, UAS), Hub genes and subgroups within the network were analyzed. In Cytoscape the "MCODE" plugin was employed to further identify core subnetworks within the PPI network. The main selection parameters were set as follows: maximum depth = 100, K-core = 2, and node score cutoff = 0.2. Additionally, the "cytoHubba" plugin in Cytoscape was

**Table 1. Main characteristics of the datasets.**

| Dataset | Sample Size | Control Samples | AR Samples | Platform | Technology | Dataset | Dataset Type |
|---------|-------------|-----------------|------------|----------|------------|---------|--------------|
| GSE75011 | 40 | 15 | 25 | GPL16791 | NexteraXT Illumina Sequencing | GSE75012 | Training Set |
| GSE43523 | 12 | 5 | 7 | GPL6883 | Illumina HumanRef-8 v3.0 Beadchip | GSE43524 | Validation Set |

used to score each node gene using five algorithms: EcCentricity, Radiality, Betweenness, Stress, and Degree. The key clusters identified by the MCODE plugin and the hub genes identified by the five algorithms of the cytoHubba plugin were combined for subsequent analysis [19].

## Lasso regression analysis

A Lasso regression model was applied to further identify the key genes with the highest predictive power [20]. The optimal hyperparameters for the Lasso regression were determined using cross-validation methods, such as the cv.glmnet function, to select the final subset of key genes.

$$Risk\ Score = \beta 1 \times mRNA1 + \beta 2 \times mRNA2 + \cdots + \beta n \times mRNAn$$

## Cell culture and treatment

Human nasal epithelial cells (HNEpC) were purchased from ATCC and cultured in DMEM medium (ThermoFisher, China) supplemented with 10% fetal bovine serum (Gibco, USA). All cells were maintained at 37 °C with 5% $CO_2$.

Initially, HNEpC cells were incubated for 24 hours in medium containing 40 µg/ml of indoor dust mite allergen (Derp1) (INDOOR Biotechnologies, USA). Following this, total RNA was extracted from the HNEpC cells for qPCR analysis to measure *PLXNB1* RNA levels. The oligonucleotide probe sequences for Plexin-B2 are as follows: Forward: 5′-CAA GCG GCG GCG GCA GAA GCG AGA-3′. Reverse: 5′-ATC TGC TGT AGG CGG AAG GCC AG-3′. The supernatants were analyzed for IL-4, IL-5, IL-6, IFNγ, and IL-10 levels. In the inhibition experiment, HNEpC cells were pretreated with *PLXNB1* siRNA, followed by RNA extraction and qPCR analysis to measure *PLXNB1* RNA levels. Supernatants were also assessed for IL-4, IL-5, IL-6, IFNγ, and IL-10. The oligonucleotide sequences for *PLXNB1* siRNA are:

Sense: 5′-AGAAGAUGCAGCUGGGCUAUU-3′, Antisense: 5′-UUCUCCGAACGUGUCACGUUU-3′

The cells were then incubated with the conditioned medium for 24 hours, with PBS serving as a control for the control group.

## Quantitative polymerase chain reaction (qPCR)

Total RNA was extracted from cultured cells using TRIzol reagent (Invitrogen, USA) according to the manufacturer's protocol. The purity and concentration of RNA were assessed using a spectrophotometer. Subsequently, RNA was reverse transcribed into complementary DNA (cDNA) using the PrimeScript RT Reagent Kit (Takara, Japan).

Quantitative PCR was performed using the Bio-Rad CFX96 real-time PCR system (Bio-Rad, USA) and SYBR Green PCR Master Mix (Qiagen, Germany). The final reaction mixture volume was 20 µL, which included 10 µL SYBR Green PCR Master Mix, 0.5 µL of each forward and reverse primer (final concentration of 0.5 µM), 1 µL of cDNA template, and 8 µL of nuclease-free water. The primers used were specifically designed for the target gene and the reference gene (GAPDH).

Relative gene expression levels were calculated using the $2^{-\Delta\Delta Ct}$ method, where ΔCt represents the difference in threshold cycle values between the target gene and the reference gene (GAPDH), and ΔΔCt represents the difference in ΔCt values between the experimental and control groups.

All experiments were performed in triplicate, and data are presented as mean±standard error of the mean (SEM).

## Animal model establishment

Twelve healthy male C57BL/6 mice (6–8 weeks old, weighing 18–22g) were purchased from were purchased from Jiangsu Jicui Yaokang Biotechnology Co., Ltd (License No. SCXK [Su] 2023−0009) and housed in a specific pathogen-free (SPF) facility under standard conditions (temperature: 22±2 °C; humidity: 55±10%; 12-hour light/dark cycle), with free access to food and water. All animal procedures strictly followed the institutional guidelines for the care and use of

laboratory animals. Animal welfare was fully respected throughout the entire experiment, and measures were taken to minimize stress, pain, and suffering. Humane treatment of animals was ensured, and cruel or inhumane behaviors were strictly prohibited. The experimental procedures adhered to the principles of the 3Rs (Replacement, Reduction, Refinement) and ensured the safety of research personnel. All methods and objectives complied with ethical standards and international conventions for animal research.

The mice were randomly assigned to two groups: a control group (n = 6) and an AR model group (n = 6).

Sensitization phase (Days 1, 5, 9, and 12): Mice in the AR group received intraperitoneal injections of 10 μg Dermatophagoides pteronyssinus allergen 1 (Der p1; Citeq Biologics B.V) mixed with 2 mg aluminum hydroxide adjuvant (Al(OH)$_3$; Sigma-Aldrich) in a total volume of 200 μL sterile phosphate-buffered saline (PBS). Control mice were injected with an equal volume of PBS.

Challenge phase (Days 13–20): Mice in the AR group were lightly anesthetized with isoflurane and intranasally administered 10 μg Der p1 (10 μL per nostril, 20 μL total in PBS) once daily for eight consecutive days. Control mice received PBS following the same protocol.

On Day 20, 30 minutes after the final intranasal challenge, allergic symptoms such as sneezing and nasal rubbing were observed and recorded. Following symptom assessment, mice were anesthetized with 2–3% isoflurane in oxygen to minimize pain and distress during sample collection. Blood samples were collected, and nasal tissues were harvested for histological and molecular analyses. Finally, mice were euthanized by $CO_2$ inhalation followed by cervical dislocation in accordance with institutional animal care guidelines. Throughout the study, animals were monitored daily for signs of distress, and any suffering was minimized following the 3Rs principles.

## Behavioral assessment and serum cytokine detection

Thirty minutes after the final intranasal Der p1 challenge on Day 19, each mouse was observed for 15 minutes to record the number of sneezing and nasal rubbing behaviors. Behavioral events were recorded manually.

On Day 20, mice were anesthetized, and blood samples were collected via retro-orbital puncture. Serum was separated by centrifugation at 3000 rpm for 10 minutes at 4 °C and stored at –80 °C. The serum levels of total IgE, IL-4, IL-6, IL-5 and TNF-α were quantified using commercial ELISA kits (Elabscience Biotechnology Co., Ltd., China), according to the manufacturer's instructions. All samples were tested in duplicate, and absorbance was measured at 450 nm using a microplate reader (Thermo Fisher Scientific, USA).

## Hematoxylin and eosin (HE) staining and histological analysis

After euthanasia, nasal tissues including the olfactory epithelium were harvested and fixed in 4% paraformaldehyde for 24 hours. Tissues were then dehydrated, embedded in paraffin, and sectioned at a thickness of 4 μm. HE staining was performed using Hematoxylin staining for 5 minutes. Differentiation in 0.5% acid-alcohol, Bluing in 0.5% ammonia water, Eosin staining for 3 minutes. Dehydration and mounting. Histopathological changes were observed under a light microscope to assess epithelial integrity and inflammatory cell infiltration.

## Quantitative real-time PCR (qRT-PCR)

Total RNA was extracted from nasal tissues using TRIzol reagent. Complementary DNA (cDNA) was synthesized using the PrimeScript RT reagent kit (Takara, Japan).

qPCR was performed using SYBR Green PCR Master Mix (Qiagen, Germany) in a 20 μL reaction system. Target genes included *PLXNB1*, IL-4, IL-5, IL-6 and TNF-α with GAPDH as the internal control. Amplification and detection were performed using a Bio-Rad CFX96 Real-Time PCR System. Relative gene expression levels were calculated using the $2^{-\Delta\Delta Ct}$ method.

## Western blot analysis

Total protein was extracted from nasal tissues using lysis buffer and quantified by the BCA method. Equal amounts of protein (approximately 30 μg) were separated by SDS-PAGE and transferred onto PVDF membranes. Membranes were blocked with 5% non-fat milk for 1 hour and incubated overnight at 4°C with the following primary antibodies: Anti-*PLXNB1* (1:1000); Anti-GAPDH (1:5000). After washing, membranes were incubated with HRP-conjugated secondary antibodies at room temperature for 1 hour. Bands were visualized using enhanced chemiluminescence (ECL) reagents, and band intensities were quantified using ImageJ software.

## Results

### DEGs and identification of olfaction-related DEGs

The study workflow is illustrated in a graphical abstract (Fig 1). We performed DEGs analysis on the GEO dataset GSE75011. The results showed a total of 1,461 DEGs between AR and HC samples. The volcano plot of DEGs between the two groups is shown in (Fig 2A), with 731 genes up-regulated and 730 genes down-regulated in the AR group. A heat-map representing the most significant DEGs is displayed in (Fig 2B). By intersecting these DEGs with 63 olfaction-related genes retrieved from the GSEA database, we identified 18 overlapping olfaction-related genes (10 upregulated and 8 downregulated) in the blood of AR patients (Fig 2C–2D).

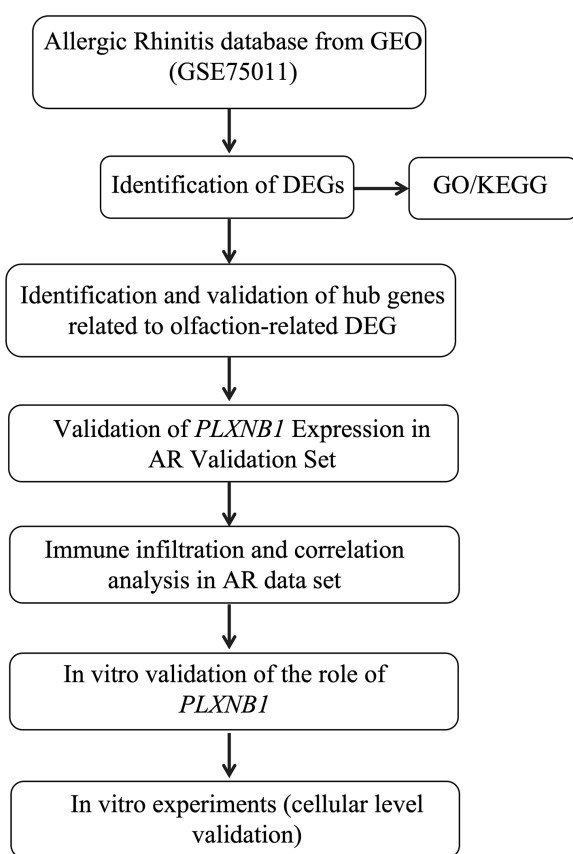

**Fig 1. The graphical abstract and workflow of this study.**

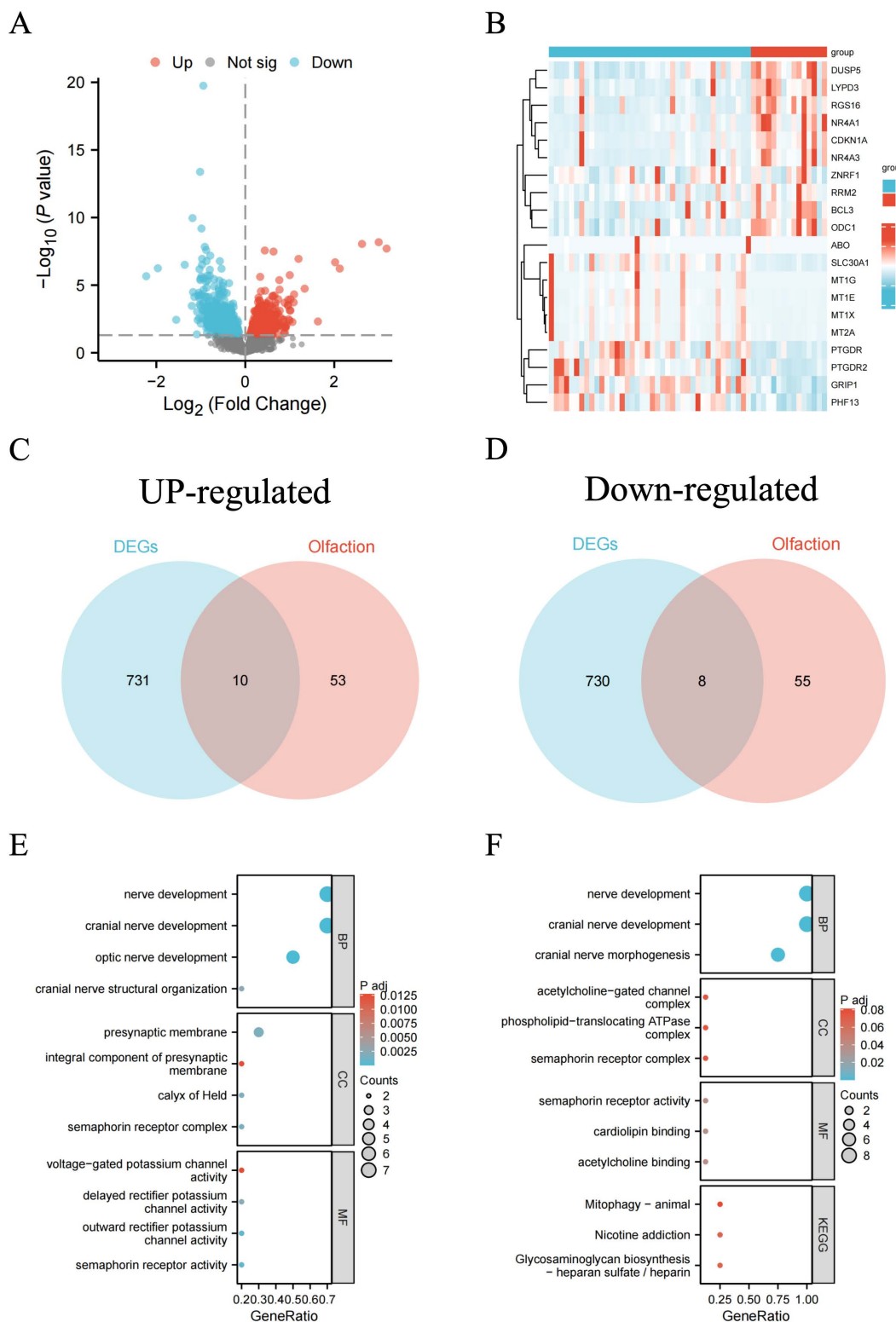

**Fig 2. Identification and functional enrichment of olfaction-related DEGs in AR. (A-B)** Volcano plot (A) and heatmap (B) of DEGs between the AR group and the control group; **(C-D)** Venn diagrams showing the intersection of olfaction-related genes from GSEA with up-regulated and down-regulated DEGs, respectively; **(E-F)** GO/KEGG enrichment analysis of up-regulated and down-regulated olfaction-related DEGs.

Next, we conducted GO/KEGG pathway enrichment analysis to explore the biological characteristics of these olfaction-related DEGs in more detail. The result show that the olfaction-related DEGs are primarily enriched in pathways related to "nerve development," "cranial nerve development," "optic nerve development," and "presynaptic membrane" (Fig 2E–2F).

### Identification of hub olfaction-DEGs from PPI analysis and machine learning

The PPI network of olfaction-related DEGs was analyzed using the STRING database and visualized with Cytoscape (Fig 3A). Important modules (gene clusters) were extracted from the PPI network using the MCODE plugin in Cytoscape. Additionally, 20 candidate genes were identified from the PPI network using five CytoHubba algorithms (Table 2). Combining these results, we obtained a total of 15 hub genes (Fig 3B–3C). These 18 candidate hub genes were then used for the final LASSO regression modeling to further narrow down the gene set (Fig 3D–3E). The results identified *CHRNB2*, *ATP8B1*, *CITED2*, *CTNNB1*, *PLXNA1*, *PLXNA3*, *SLC25A46*, *PLXNB1,* as the key hub olfaction-related DEG through LASSO regression. Finally, by intersecting the key genes identified by Lasso regression with those selected by Cytoscape, we identified *PLXNA1*, *PLXNA3*, and *PLXNB1* as the key hub olfaction-related DEGs (Fig 4A–4B).

### Validation of *PLXNB1* expression

Compared to HC samples, the expression levels of *PLXNA1*, *PLXNA3*, and *PLXNB1* were significantly lower in AR lesion and non-lesion samples (Fig 4C–4E). This trend was also observed in the validation set (GSE43523) (Fig 4F–4H). The risk score for each patient's gene expression was calculated using the following algorithm:

Risk Score = -0.02* *PLXNB1* + 0.003* *PLXNA*1 + 0.004* *PLXNA3*

As the expression level of *PLXNB1* increases, the risk score decreases; conversely, as the expression level of *PLXNB1* decreases, the risk score increases. Higher expressions of *PLXNA1* and P*LXNA3* are associated with higher risk scores, indicating that they may be linked to an increased risk of the outcome.

Since *PLXNA1* did not show statistical significance in the validation set, and we observed a more pronounced trend in the changes of *PLXNB1*, combined with the literature indicating a close relationship between *PLXNB1* and olfactory effects in AR, we decided to select *PLXNB1* for further experimental validation.

In AR, immune cell infiltration and the association of olfaction-related DEGs with differential immune cells

Through ssGSEA, we can better understand differences in immune function. The ssGSEA analysis showed significant enrichment of CD8 T cells, NK CD56 bright cells, T cells, TFH Th1 cells, and TReg in the AR group (Fig 5A–5B). Furthermore, we explored the correlation of *PLXNB1* with immune cells (Fig 5C). The results indicate that *PLXNB1* is negatively correlated with CD8 T cells, NK CD56bright cells, and TReg, but positively correlated with innate and adaptive immune cells such as Tfh cells. These findings suggest that the hub gene *PLXNB1* may reflect immune cell infiltration in AR patients.

### Functional annotation of *PLXNB1*

The top 20 genes most relevant to *PLXNB1* including *ARHGEF12*, *ARHGEF11*, *SEMA4D*, *SEMA4A*, *RRAS, RND1*, *ERBB2*, *MET*, *RAC2*, and *RAC3* (Fig 6A). GO analysis revealed that *PLXNB1*and its associated genes were predominantly enriched in regulation of cell morphogenesis, regulation of cell shape, GTPase activity, Axon guidance and MAPK signaling pathway (Fig 6B). These findings may provide clues to elucidate the role of *PLXNB1* in AR.

### Down-regulation of *PLXNB1* in AR model mice and its association with inflammatory responses

Compared to the control group, mice in the AR group showed a significant increase in the number of sneezes and nasal rubbing events within 15 minutes (P<0.0001), along with a markedly elevated serum IgE concentration (P<0.0001),

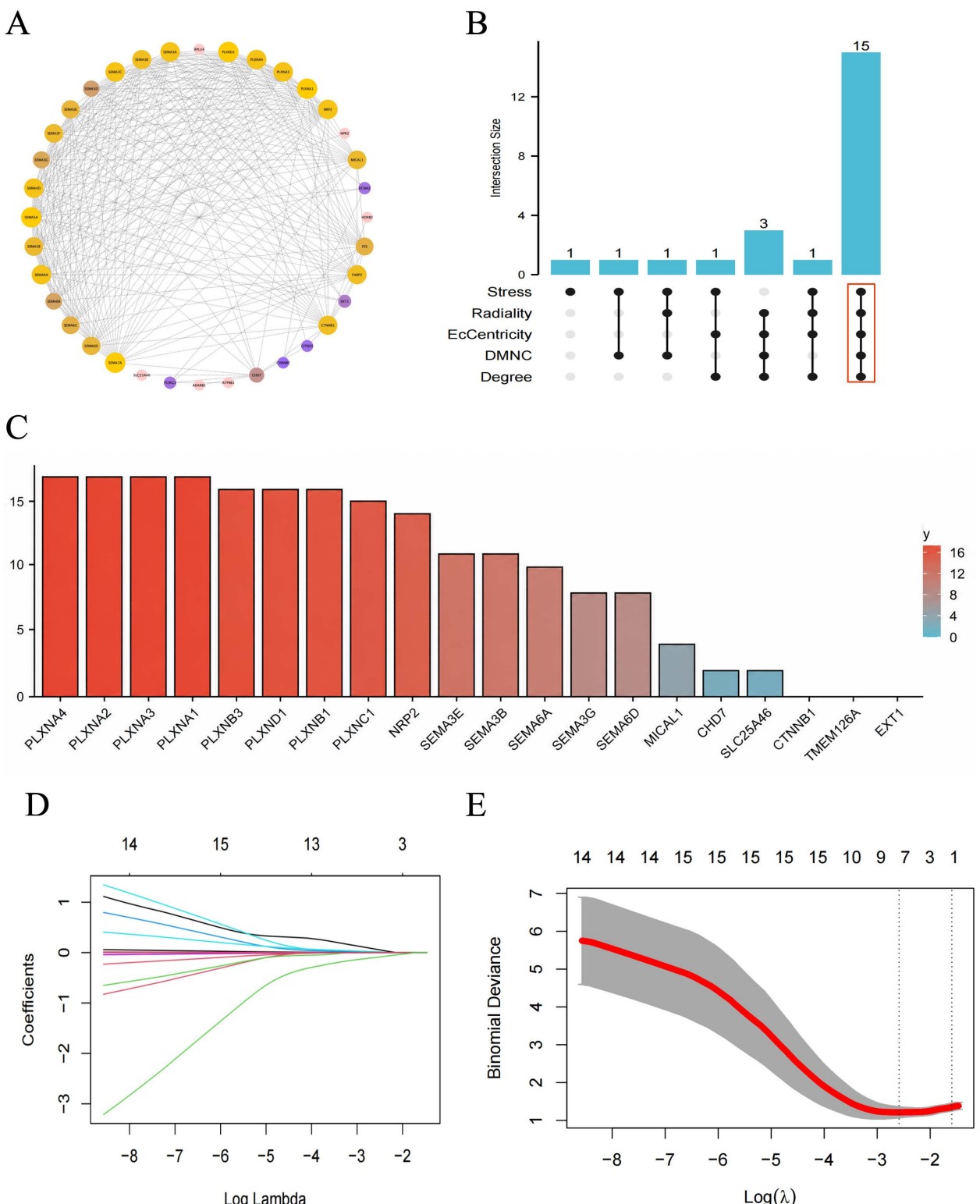

**Fig 3. Identification of Potential Hub Genes. (A)** PPI network of Olfaction-DEs. **(B)** Venn diagram depicting overlapping key gene variables from PPI analysis.(C) Visualize the Scores of Common Genes Identified by MCODE and CytoHubba. **(D-E)** Selection of candidate hub Olfaction-DEs using LASSO regression.

**Table 2. Gene list ranked by five centrality algorithms.**

| Radiality | Stress | EcCentricity | DMNC | Degree |
|---|---|---|---|---|
| PLXNA4 | SEMA3E | SEMA3E | SEMA3E | PLXNA4 |
| PLXNA2 | SEMA3B | PLXNB3 | SEMA6A | PLXNA2 |
| PLXNA3 | CHD7 | SEMA6A | SEMA3B | PLXNA3 |
| PLXNA1 | PLXNA4 | PLXNA4 | SEMA3F | PLXNA1 |
| PLXNB3 | PLXNA2 | PLXNC1 | SEMA6B | PLXNB3 |
| PLXND1 | PLXNA3 | PLXNA2 | NRP2 | PLXND1 |
| PLXNB1 | PLXNA1 | PLXND1 | PLXNC1 | PLXNB1 |
| PLXNC1 | SLC25A46 | PLXNA3 | SEMA3G | PLXNC1 |
| NRP2 | PLXNB3 | PLXNB1 | SEMA6D | NRP2 |
| SEMA3E | PLXND1 | PLXNA1 | SEMA6C | SEMA3E |
| SEMA6A | PLXNB1 | NRP2 | PLXNB3 | SEMA3B |
| SEMA3B | PLXNC1 | CHD7 | PLXND1 | SEMA6A |
| SEMA3F | NRP2 | SEMA3G | PLXNB1 | SEMA3F |
| SEMA6B | SEMA3G | MICAL1 | PLXNA4 | SEMA6B |
| SEMA3G | MICAL1 | SEMA6D | PLXNA2 | SEMA3G |
| SEMA6D | SEMA6D | SEMA3B | PLXNA3 | SEMA6D |
| SEMA6C | SEMA6A | SEMA6C | PLXNA1 | SEMA6C |
| MICAL1 | CTNNB1 | SEMA3F | MICAL1 | MICAL1 |
| CHD7 | TMEM126A | SEMA6B | CTNNB1 | CHD7 |
| CTNNB1 | EXT1 | SLC25A46 | TMEM126A | SLC25A46 |

indicating successful establishment of the model with typical allergic responses (Fig 7A–7C). ELISA results showed that the serum levels of IL-5, IL-4, IL-6, and TNF-α in the AR group were significantly higher than those in the control group (Fig 7D–G), suggesting that the model exhibited systemic immune activation and inflammatory responses.

HE staining showed that the nasal epithelium of control mice was structurally intact and well organized (Fig 8A). In contrast, the AR group exhibited disorganized epithelial cells with local shedding and marked inflammatory cell infiltration in the lamina propria (Fig 8B). In nasal tissues, qPCR analysis revealed that *PLXNB1* mRNA expression was significantly lower in the AR group compared to the control group (Fig 8C). Western blot analysis further confirmed the downregulation of *PLXNB1* protein in the AR group (Fig 8D). Meanwhile, the mRNA expression levels of inflammatory cytokines such as IL-4, IL-6, and TNF-α were markedly elevated, showing a negative correlation with *PLXNB1* expression.

### During allergen stimulation, the nasal mucosal epithelium mediates inflammation through *PLXNB1*, leading to olfactory dysfunction

During allergen stimulation, nasal mucosal epithelium mediates inflammation through *PLXNB1*, leading to OD. *PLXNB1* is indeed a key factor in the development of AR with OD, although the mechanisms mediating this process remain unclear. We constructed a simplified in vitro AR model by stimulating the HNEpC cell line with Derp1. The results showed that *PLXNB1* mRNA levels were significantly lower in the Derp1-treated HNEpC cells compared to the control group ($P < 0.01$) (Fig 9A). ELISA results showed that IgE levels were significantly in the experimental group ($P < 0.001$), indicating successful construction of the in vitro AR cell model (Fig 9B) Consequently, we investigated the functional role of *PLXNB1* in the AR model by utilizing siRNA transfection to suppress *PLXNB1* expression. The efficacy of transfection was confirmed at the mRNA level through qPCR (Fig 9C). ELISA results showed that IL-4, IL-5, and IL-6 levels were significantly elevated in the supernatants of si-*PLXNB1* treated cell models ($P < 0.001$) (Fig 9D–9F).

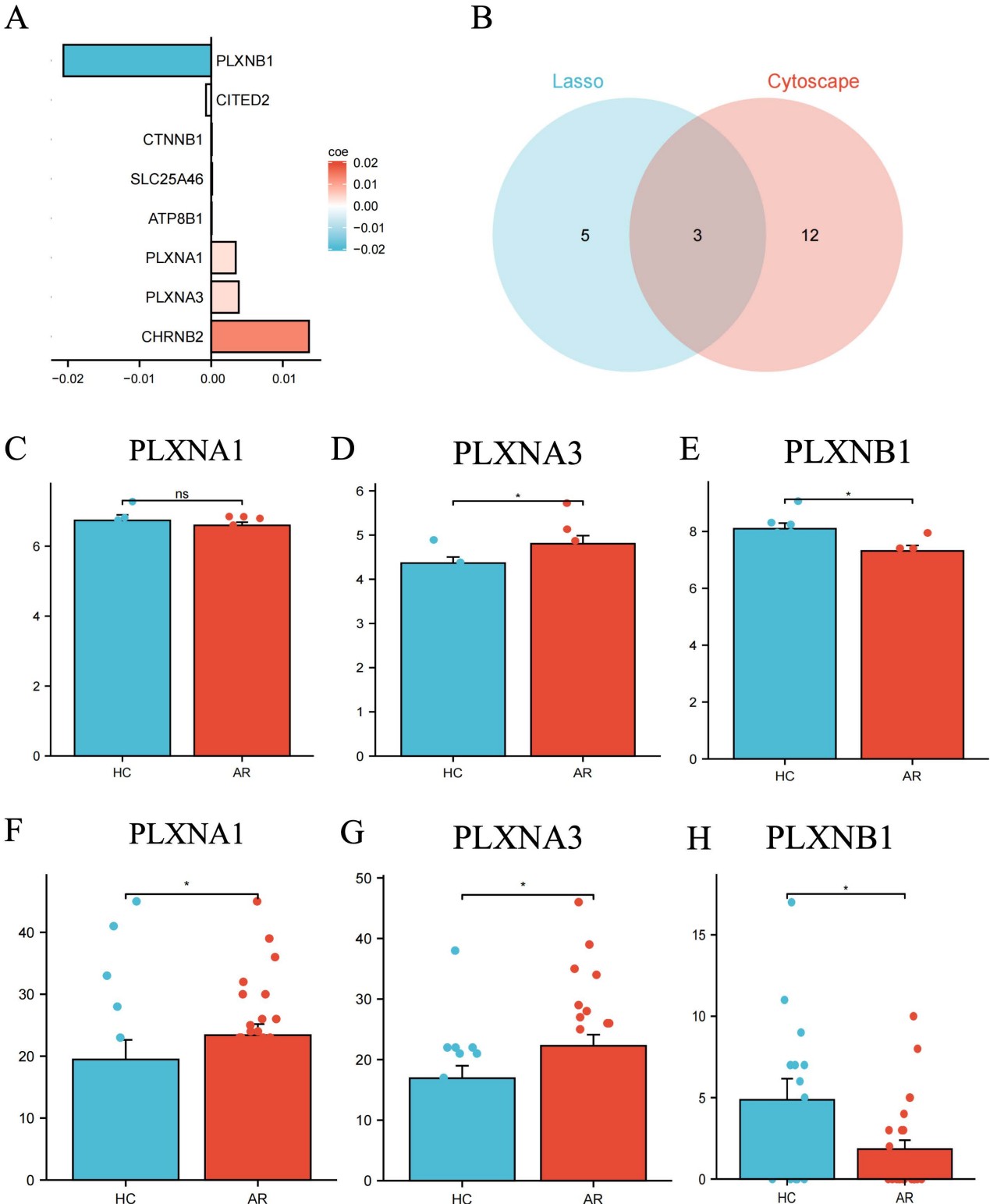

**Fig 4. Olfaction-related DEGs level in AR. (A)** Parameters of variables in the Lasso regression model; **(B)** LASSO and PPI network analysis identified *PLXNB1*, *PLXNA1* and *PLXNA3* as jointly determined predictive factors for AR **(C-E)** *PLXNB1*, *PLXNA1* and *PLXNA3* expression level in GSE75011. **(F-H)** *PLXNB1*, *PLXNA1* and *PLXNA3* expression level in GSE43523.

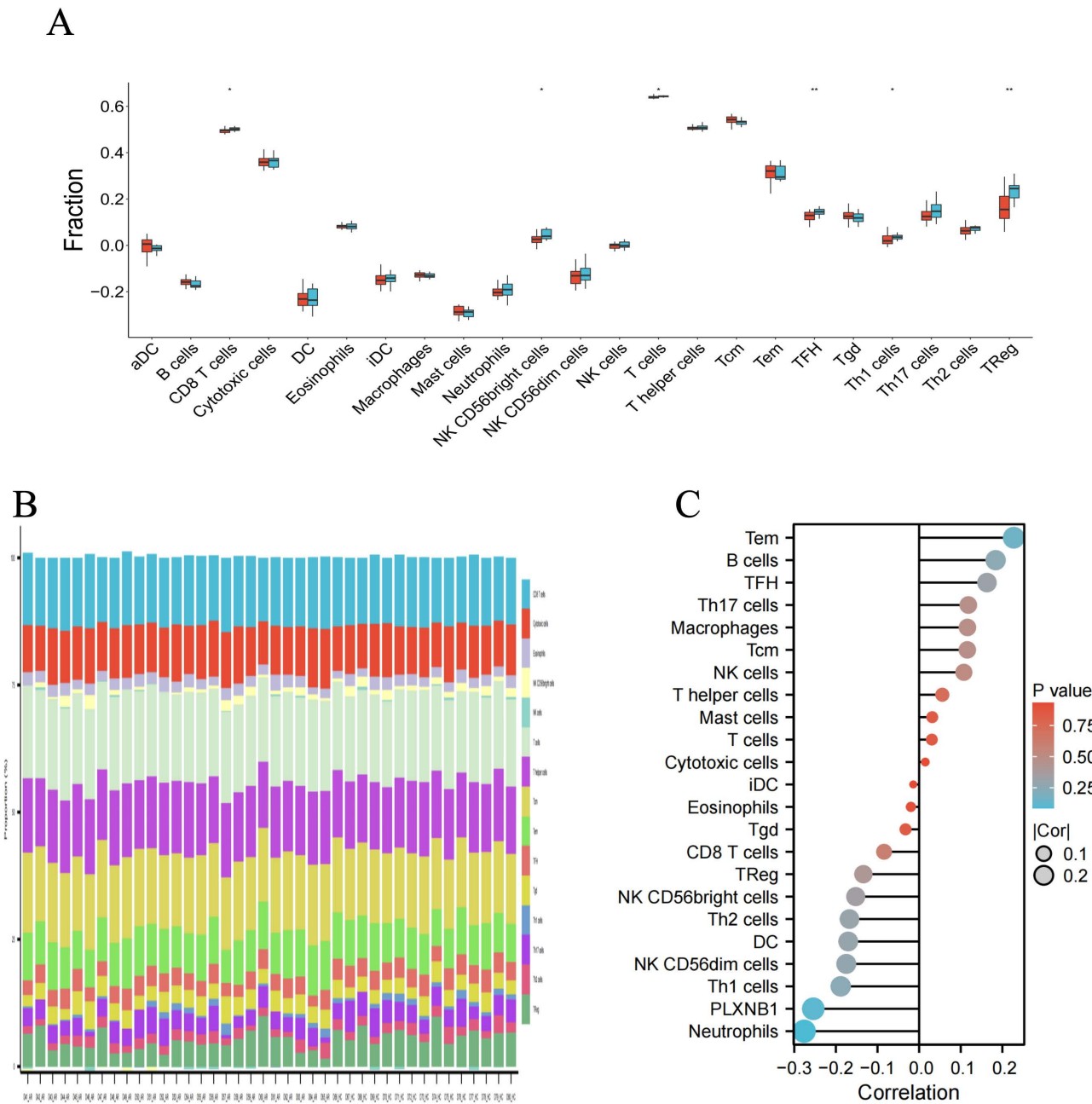

**Fig 5. Examination of immune cell infiltration in AR patients and the correlation between hub genes and different immune cells. (A)** Violin plots showing the expression of immune cells in AR and HC tissues. **(B)** Bar plot of the accumulation of immune cells. **(C)** Correlation between *PLXNB1* and immune infiltrating cells.

### *PLXNB1* modulates inflammatory signaling pathways in AR models

To further explore the regulatory role of *PLXNB1* in allergic inflammation, we manipulated its expression in Derp1-stimulated HNEpC cells. *PLXNB1* knockdown using siRNA significantly increased the expression of pro-inflammatory cytokines IL-4, IL-5, and IL-6 (P < 0.01), indicating that *PLXNB1* may act as a negative regulator of inflammation. Western

A

B

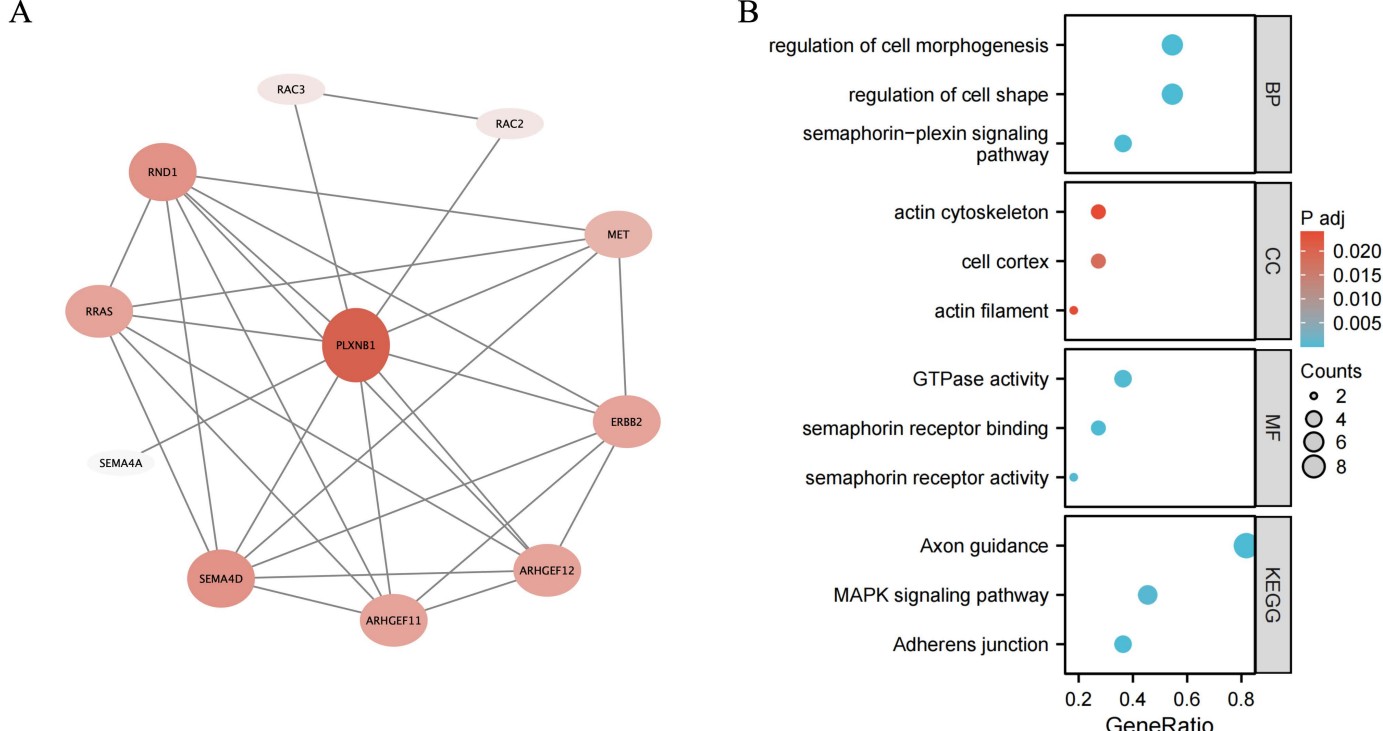

**Fig 6. Functional annotation of *PLXNB1*. (A)** The Top 10 genes associated with *PLXNB1*; **(B)** GO/KEGG functional annotation of the associated genes of *PLXNB1*.

blot analysis showed elevated phosphorylation of p38 MAPK and upregulated TNF-α, suggesting activation of the p38 pathway contributes to the enhanced inflammatory response (Fig 10A).

Conversely, overexpression of *PLXNB1* (ov-*PLXNB1*) in Derp1-treated cells resulted in a significant reduction in IL-4, IL-6, and TNF-α levels, accompanied by decreased p-p38 expression (P < 0.01), suggesting that *PLXNB1* suppresses inflammation via inhibition of the *MAPK/p38* signaling pathway (Fig 10B).

Furthermore, treatment with the antihistamine Desloratadine partially restored *PLXNB1* expression and significantly reduced inflammatory cytokine expression at both the mRNA and protein levels (P < 0.05) (Fig 10C). These findings support that pharmacological modulation of *PLXNB1* may alleviate AR-associated inflammation and potentially improve olfactory dysfunction (Fig 10D).

## Discussion

This study revealed olfaction-related genes associated with AR through bioinformatics analysis. Initially, we retrieved and analyzed datasets (GSE75011) from public databases, confirming a set of DEGs associated with OD in AR patients. These Olfaction-DEGs were predominantly enriched in biological processes related to nerve development and olfactory signal transduction, such as "nerve development" and "presynaptic membrane". Further PPI network analysis and machine learning screening identified *PLXNB1* as a pivotal Olfaction-DEG. *PLXNB1* exhibited significantly lower expression in AR patients compared to healthy controls, suggesting its potential association with OD. Application of the LASSO regression model further validated the significance of *PLXNB1* in AR, with its expression levels negatively correlated with disease risk scores.

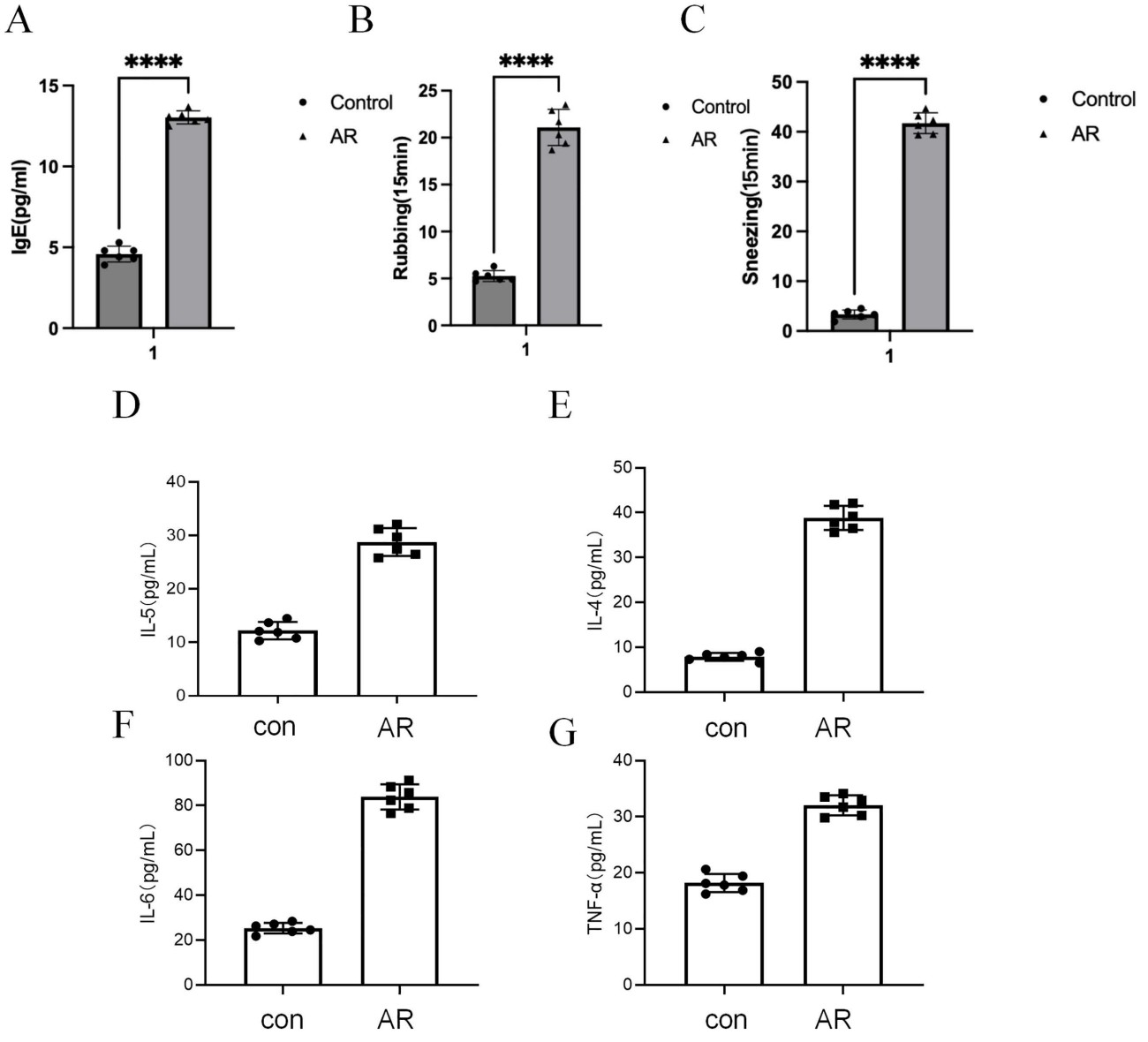

**Fig 7. Behavioral symptoms and serum inflammatory markers in the AR mouse model. (A)** Serum IgE levels were significantly elevated in the AR group compared to the control group. **(B)** Number of nasal rubbing events within 15 minutes. **(C)** Number of sneezes within 15 minutes. **(D–G)** Serum concentrations of IL-5, IL-4, IL-6, and TNF-α were significantly increased in AR mice, as measured by ELISA. Data are presented as mean±SEM, n=6 per group. ****P<0.0001 (unpaired two-tailed Student's t-test).

*PLXNB1* belongs to the family of transmembrane receptors and has been implicated in various physiological processes, including axon guidance, cell migration, and regulation of immune responses [21]. Previous studies have shown that allergens and vascular endothelial growth factor enhance the expression of neuroimmune semaphorins 4A (*Sema4A*) and 4D (*Sema4D*) and their receptors in the lungs [22–24], where *PLXNB1* acts as a receptor for *Sema4A* and *Sema4D*, playing a role in allergic airway inflammation [25,26]. Furthermore, *PLXNB1* is known to modulate immune responses through interactions with its ligands, affecting the migration and activation of immune cells [27]. Chapoval SP et al. have demonstrated that the absence of *PLXNB1* in vivo results in exacerbated allergic airway inflammation and mucin

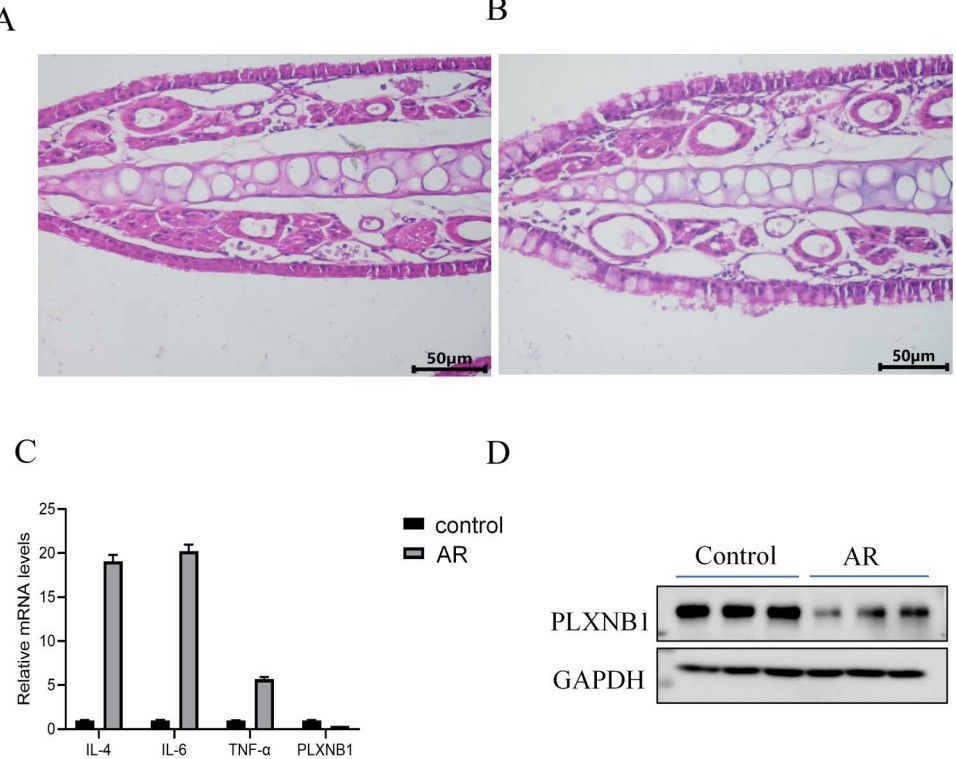

**Fig 8. Histological changes and *PLXNB1* expression in nasal tissues of AR model mice. (A)** HE staining of nasal mucosa in control mice showing intact epithelial structure and orderly cell arrangement. **(B)** HE staining of AR mice showing disorganized epithelium, epithelial shedding, and prominent inflammatory cell infiltration in the lamina propria. (C) qPCR analysis showing significantly reduced *PLXNB1* mRNA expression in AR mice compared to controls. **(D)** Western blot analysis confirming the downregulation of *PLXNB1* protein in AR mice. Increased mRNA expression of inflammatory cytokines IL-4, IL-6, and TNF-α was observed, showing an inverse correlation with *PLXNB1* levels.

dysregulation in response to allergen exposure. In AR, immune dysregulation plays a critical role in the chronic inflammatory response triggered by allergens [28]. Changes in *PLXNB1* expression may influence the behavior of immune cells in the nasal mucosa, potentially exacerbating inflammation and affecting the severity of the disease.

*PLXNB1* plays a crucial role in regulating immune infiltration and olfactory nerve function in AR. Acting as a receptor for *Sema4A* and *Sema4D*, *PLXNB1* is involved in modulating the migration and activation of immune cells, which are essential for regulating inflammatory responses in the nasal cavity. Abnormal activation and aggregation of immune cells in AR lead to chronic inflammation of the nasal mucosa, thereby affecting olfactory nerve function. In our study, the ssGSEA analysis showed significant enrichment of CD8 T cells, NK CD56 bright cells, T cells, TFH Th1 cells, and TReg in the AR group. Consistent with previous studies, *PLXNB1* is a key regulator of the stability and function of human Treg cells, The results indicate that *PLXNB1* is negatively correlated with CD8 T cells, NK CD56 bright cells, and TReg, but positively correlated with innate and adaptive immune cells such as Tfh cells. In AR, immune dysregulation plays a critical role in the chronic inflammatory response triggered by allergens. Changes in *PLXNB1* expression may influence the behavior of immune cells in the nasal mucosa, potentially exacerbating inflammation and affecting the severity of the disease [29].

Studies indicate that changes in *PLXNB1* expression directly influence the behavior of immune cells in the nasal cavity [30,31]. Under allergen stimulation, decreased *PLXNB1* expression may enhance the migration and aggregation of immune cells, exacerbating nasal inflammation. This inflammatory response not only affects nasal mucosal tissues but may also directly impact normal olfactory nerve function through neuro-immune signaling pathways. OD is prevalent in

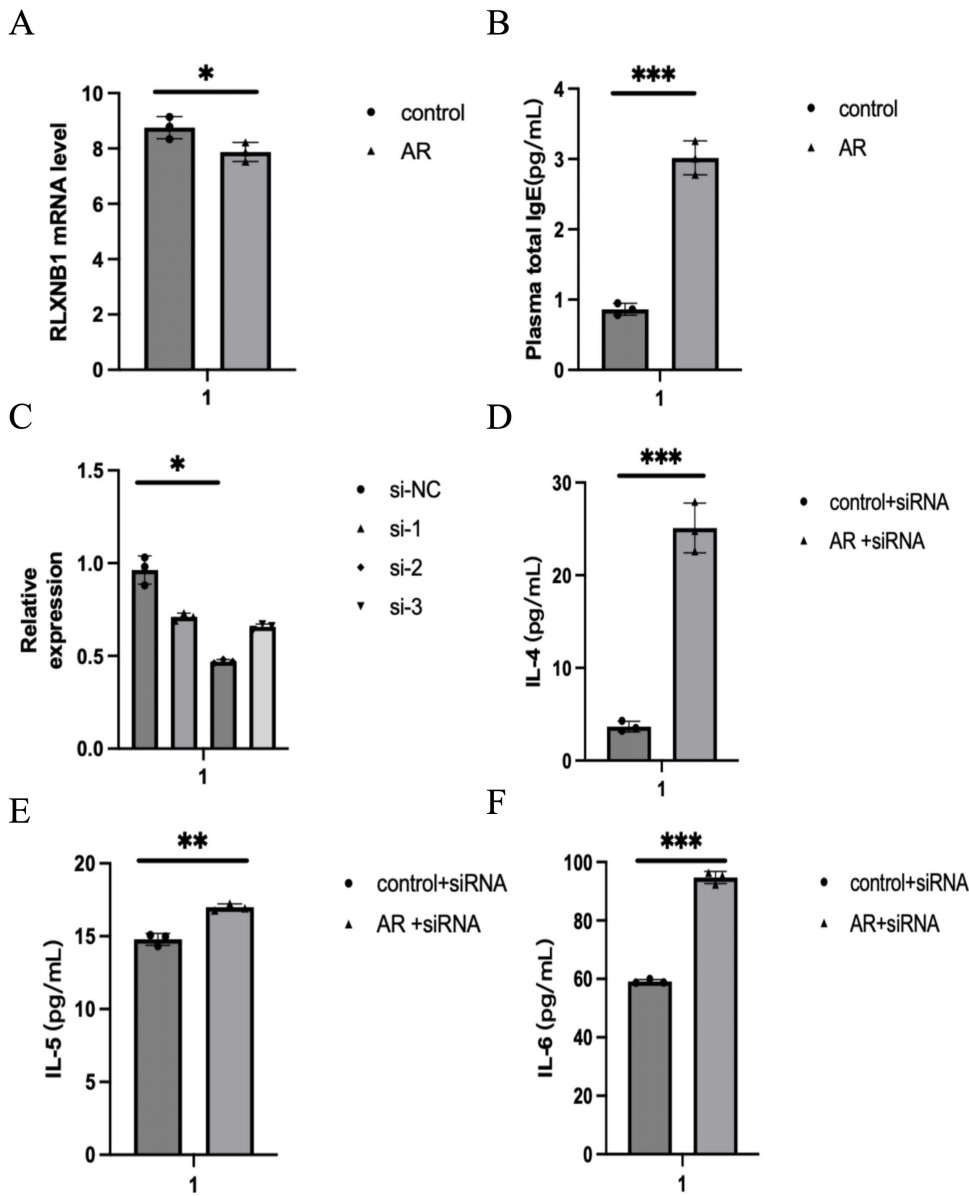

**Fig 9. *PLXNB1* levels in AR are closely related to inflammation in the OB. (A)** mRNA expression levels of *PLXNB1* in cells cultured from the control group and Derp1-treated group; **(B)** Levels of IgG in the supernatant of cells cultured from the control group and Derp1-treated group; **(C)** Assessment of *PLXNB1* expression in HNEpC cells and transfection efficiency of siRNA by RT-qPCR; **(D)** IL-4 expression levels in the serum of the Control + siRNA group and Derp1 + siRNA treatment group; **(E)** IL-5 expression levels in the serum of the Control + siRNA group and Derp1 + siRNA treatment group; **(F)** IL-6 expression levels in the serum of the Control + siRNA group and Derp1 + siRNA treatment group.

AR patients and is closely associated with neural damage and impaired olfactory perception related to inflammation. We observed that the deletion of the *PLXNB1* gene leads to tissue inflammation, mucus cell hyperplasia, and increased local production of IL-4 and IL-6. IL-4 and IL-5 are pro-inflammatory cytokines associated with Th2-type immune responses [32]. Their up-regulation suggests that in the absence of *PLXNB1*, Th2-type immune responses may be enhanced, which is associated with diseases such as allergies and asthma [28,29]. This indicates that *PLXNB1* may help maintain

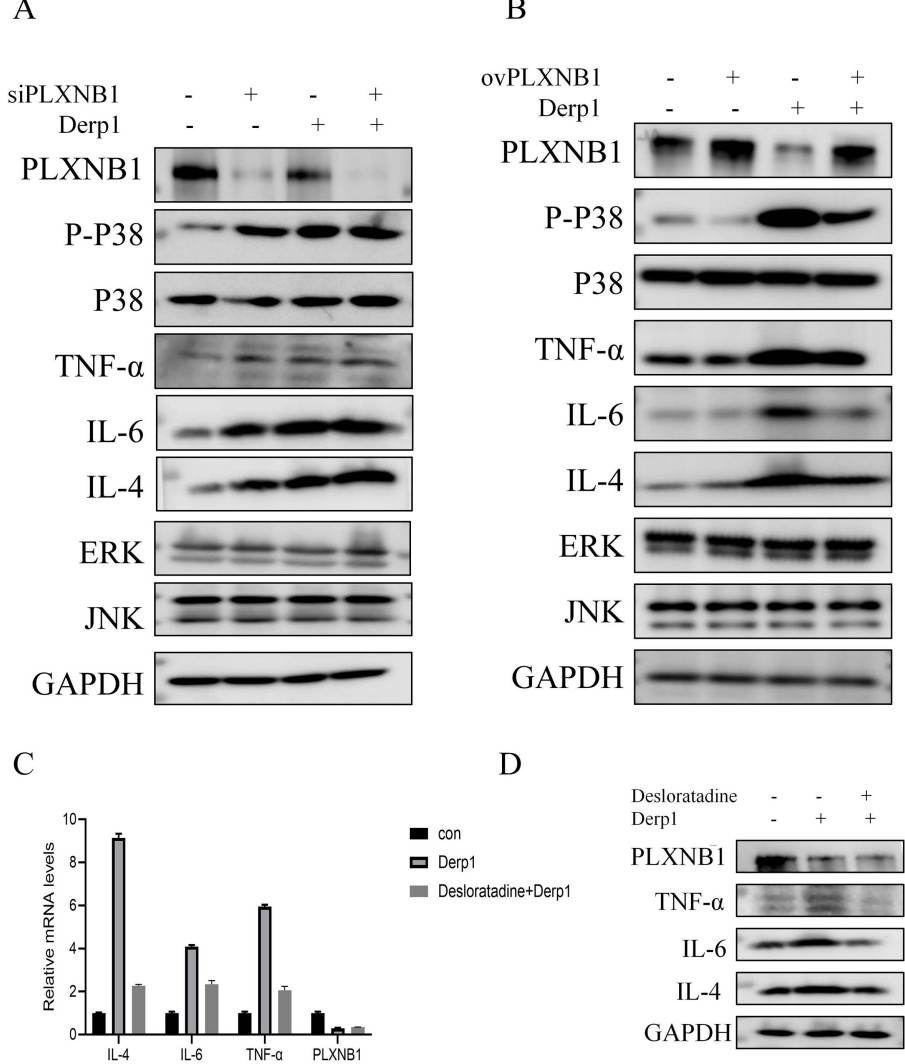

**Fig 10. _PLXNB1_ regulates inflammatory cytokine expression and MAPK signaling pathway in AR models. (A)** Western blot analysis showed that siRNA-mediated knockdown of _PLXNB1_ under Derp1 stimulation increased the expression of p-P38, TNF-α, IL-6, and IL-4, indicating activation of inflammatory pathways; **(B)** Overexpression of _PLXNB1_ suppressed Derp1-induced inflammation, as evidenced by reduced protein levels of p-P38, TNF-α, IL-6, and IL-4; (C) qPCR results demonstrated that Desloratadine treatment decreased Derp1-induced mRNA expression of IL-4, IL-6, and TNF-α, and partially restored _PLXNB1_ expression; **(D)** Western blot further confirmed that Desloratadine reduced the protein levels of inflammatory cytokines (TNF-α, IL-6, IL-4) and increased _PLXNB1_ expression.

a balanced immune response under normal conditions by inhibiting excessive pro-inflammatory reactions. This indicates that_PLXNB1_ may help maintain a balanced immune response under normal conditions by inhibiting excessive pro-inflammatory reactions.

This study demonstrates that _PLXNB1_ is a key gene involved in the development of OD associated with AR. Through integrative bioinformatics analysis, animal model validation, and in vitro cellular experiments, we found that _PLXNB1_ expression was significantly downregulated in AR models, which correlated closely with the upregulation of inflammatory cytokines such as IL-4, IL-6, and TNF-α, as well as the activation of the MAPK/p38 signaling pathway. Further experiments involving _PLXNB1_ over-expression and knockdown revealed that _PLXNB1_ acts as a negative regulator of

inflammation. Moreover, treatment with the antihistamine Desloratadine partially restored *PLXNB1* expression and alleviated the inflammatory response, suggesting that Desloratadine may exert its therapeutic effects through a *PLXNB1*-related mechanism.

In recent years, studies on olfactory dysfunction have indicated that it is not solely caused by physical factors such as nasal obstruction or increased secretions, but may also involve the development, injury, and regeneration of olfactory epithelial neurons, which are influenced by inflammatory mediators [33,34]. *PLXNB1*, a known axon guidance receptor, participates in diverse biological processes including cell migration, apoptosis, and inflammation regulation. Previous research has shown that *PLXNB1* can regulate *RhoA/ROCK* and MAPK signaling pathways, thereby influencing cellular morphology and immune responses. Our study is the first to demonstrate in an AR-related OD model that *PLXNB1* down-regulation is associated with p38 pathway activation, suggesting that *PLXNB1* may influence olfactory function by modulating local inflammatory responses.

Additionally, the p38 MAPK pathway, a key player in chronic inflammation and tissue damage, has been implicated in the release of inflammatory cytokines and airway remodeling in various respiratory diseases. In our study, siRNA-mediated *PLXNB1* knockdown led to a significant increase in p-p38 levels, indicating that *PLXNB1* may suppress inflammation by inhibiting this pathway. Importantly, we observed that Desloratadine not only reduced the expression of inflammatory cytokines but also upregulated *PLXNB1* expression, implying that *PLXNB1* may represent a previously unrecognized therapeutic target of traditional antihistamines. This finding expands our understanding of the pharmacological mechanisms of anti-allergic drugs and provides a theoretical basis for future targeted interventions in AR-associated olfactory dysfunction.

Although this study systematically combined bioinformatics analysis, in vitro cellular experiments, and in vivo animal experiments to explore the role of *PLXNB1* in allergic rhinitis and olfactory dysfunction, certain limitations remain. The datasets obtained from the public GEO database had relatively small sample sizes, which may introduce bias and affect the generalizability and robustness of the results. Future studies with larger cohorts and multi-center clinical data are needed for further validation. Although our findings suggest that *PLXNB1* may mediate inflammatory responses through the *MAPK/p38* signaling pathway, its upstream regulators and downstream targets have not been fully elucidated and require further investigation. While the in vitro and in vivo experiments provide preliminary evidence, direct validation in clinical patient samples (e.g., nasal mucosal tissues, peripheral blood) is still lacking, and thus more evidence is required to support its translational value. In addition, in the cellular experiments, Der p1 stimulation was used to mimic allergen exposure; however, other environmental and genetic factors contributing to the pathogenesis of AR were not taken into account.

## Conclusion

Through the integration of bioinformatics analysis, in vitro cellular experiments, and in vivo animal experiments, this study systematically revealed the potential role of *PLXNB1* in AR and its associated olfactory dysfunction. The results demonstrated that *PLXNB1* was significantly downregulated in both AR patients and experimental models, and its reduced expression was closely associated with the up-regulation of inflammatory cytokines, potentially mediating inflammatory responses via activation of the *MAPK/p38* signaling pathway. In contrast, overexpression of *PLXNB1* or intervention with the antihistamine Desloratadine partially restored *PLXNB1* expression and suppressed inflammation, suggesting a protective role of *PLXNB1* in disease regulation.

## Supporting information

**S1. Raw_images.**
(PDF)

   

## Acknowledgments

We would like to thank The Second Affiliated Hospital of Chongqing Medical University for providing technical support and excellent conditions for sample collection, experimental procedures, and data analysis. We also sincerely appreciate our laboratory colleagues for their valuable suggestions and assistance throughout the research process.

Special thanks are extended to all the teachers and peer experts who offered guidance and support in the study design, experimental implementation, and manuscript preparation, which greatly contributed to the improvement of this work.

## Author contributions

**Data curation:** Xinglong Chen, Lingqiong Zhao.

**Formal analysis:** Lingqiong Zhao.

**Supervision:** Lingqiong Zhao.

**Validation:** Lingqiong Zhao.

**Writing – original draft:** Xinglong Chen.

**Writing – review & editing:** Wenlong Luo.

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
