## [Decision Letter · Decision Letter 0]

12 Mar 2026

PONE-D-25-63192Study on the Pathogenesis of PLXNB1 Gene in Olfactory Dysfunction of Allergic RhinitisPLOS One

Dear Dr. wenlong,

Thank you for submitting your manuscript to PLOS ONE. After careful consideration, we feel that it has merit but does not fully meet PLOS ONE’s publication criteria as it currently stands. Therefore, we invite you to submit a revised version of the manuscript that addresses the points raised during the review process.

We look forward to receiving your revised manuscript.

Kind regards,

Ali Faramarzi, MD, MPH

Academic Editor

PLOS One

Journal Requirements:

2. To comply with PLOS One submissions requirements, in your Methods section, please provide additional information regarding the experiments involving animals and ensure you have included details on (1) methods of sacrifice, (2) methods of anesthesia and/or analgesia, and (3) efforts to alleviate suffering.

4. In the online submission form you indicate that your data is not available for proprietary reasons and have provided a contact point for accessing this data. Please note that your current contact point is a co-author on this manuscript. According to our Data Policy, the contact point must not be an author on the manuscript and must be an institutional contact, ideally not an individual. Please revise your data statement to a non-author institutional point of contact, such as a data access or ethics committee, and send this to us via return email. Please also include contact information for the third party organization, and please include the full citation of where the data can be found.

6. Please ensure that you refer to Figure 1 in your text as, if accepted, production will need this reference to link the reader to the figure.

7. PLOS ONE now requires that authors provide the original uncropped and unadjusted images underlying all blot or gel results reported in a submission’s figures or Supporting Information files. This policy and the journal’s other requirements for blot/gel reporting and figure preparation are described in detail at https://journals.plos.org/plosone/s/figures#loc-blot-and-gel-reporting-requirements and https://journals.plos.org/plosone/s/figures#loc-preparing-figures-from-image-files. When you submit your revised manuscript, please ensure that your figures adhere fully to these guidelines and provide the original underlying images for all blot or gel data reported in your submission. See the following link for instructions on providing the original image data: https://journals.plos.org/plosone/s/figures#loc-original-images-for-blots-and-gels.

Reviewers' comments:

Reviewer's Responses to Questions

**Comments to the Author**

1. Is the manuscript technically sound, and do the data support the conclusions?

Reviewer #1: Yes

2. Has the statistical analysis been performed appropriately and rigorously? 

Reviewer #1: Yes

3. Have the authors made all data underlying the findings in their manuscript fully available?

Reviewer #1: Yes

4. Is the manuscript presented in an intelligible fashion and written in standard English?

Reviewer #1: Yes

5. Review Comments to the Author

Reviewer #1: 1. Although the importance of the PLXNB1 gene in the nervous system has been partially revealed by previous studies, its specific role in olfactory dysfunction in patients with allergic rhinitis (AR) has not been fully investigated.

2. It is the first time to systematically explore the expression characteristics and potential mechanisms of the PLXNB1 gene in olfactory dysfunction caused by AR in this study.

3. During the development of AR, the nasal mucosa is stimulated by allergens, triggering an immune inflammatory response that damages olfactory neurons. The Plexin B1 protein encoded by the PLXNB1 gene plays a key role in neuronal development and repair. This gene may be regulated during inflammation, with downregulation or deficiency exacerbating olfactory dysfunction.

4. The interaction between the inflammatory response and changes in PLXNB1 function may create a vicious cycle, further worsening the reduction in odor perception, impairment of odor discrimination ability, and overall decline in olfactory quality. This mechanism suggests the potential value of PLXNB1 as a therapeutic target.

5. The p38 MAPK pathway, a key player in chronic inflammation and tissue damage, has been implicated in the release of inflammatory cytokines and airway remodeling in various respiratory diseases. In our study, siRNA-mediated PLXNB1 knockdown led to a significant increase in p-p38 levels, indicating that PLXNB1 may suppress inflammation by inhibiting this pathway. Importantly, we observed that Desloratadine not only reduced the expression of inflammatory cytokines but also upregulated PLXNB1 expression, implying that PLXNB1 may represent a previously unrecognized therapeutic target of traditional antihistamines. This finding expands our understanding of the pharmacological mechanisms of anti-allergic drugs and provides a theoretical basis for future targeted interventions in AR-associated olfactory dysfunction.

6. PLOS authors have the option to publish the peer review history of their article (what does this mean?). If published, this will include your full peer review and any attached files.

Reviewer #1: No

---

## [Author Response · Author response to Decision Letter 1]

23 Mar 2026

Response to Reviewer #1 We sincerely thank the reviewer for the careful evaluation of our manuscript and for the positive and constructive comments. We appreciate the reviewer’s recognition of the novelty and significance of our study. The reviewer’s comments mainly summarize the background, significance, and key findings of our work. As no specific revisions were requested, we have carefully checked the manuscript again and made minor language polishing to improve clarity. We are grateful for the reviewer’s insightful evaluation and support for our work. Thank you for your time and consideration.

Dear Editor, Thank you for your valuable comments. We have carefully revised the manuscript according to your suggestion. Specifically, additional details regarding animal experiments have been incorporated into the Methods section under “Animal model establishment,” including information on anesthesia, euthanasia, and measures taken to minimize animal suffering. We believe these revisions fully address your concerns. Thank you again for your consideration.

---

## [Decision Letter · Decision Letter 1]

11 May 2026

Study on the Pathogenesis of PLXNB1 Gene in Olfactory Dysfunction of Allergic Rhinitis

PONE-D-25-63192R1

Dear Dr. wenlong,

We’re pleased to inform you that your manuscript has been judged scientifically suitable for publication and will be formally accepted for publication once it meets all outstanding technical requirements.

Kind regards,

Ali Faramarzi, MD, MPH

Academic Editor

PLOS One

Additional Editor Comments (optional):

Reviewers' comments:

Reviewer's Responses to Questions

**Comments to the Author**

1. If the authors have adequately addressed your comments raised in a previous round of review and you feel that this manuscript is now acceptable for publication, you may indicate that here to bypass the “Comments to the Author” section, enter your conflict of interest statement in the “Confidential to Editor” section, and submit your "Accept" recommendation.

Reviewer #1: All comments have been addressed

2. Is the manuscript technically sound, and do the data support the conclusions?

Reviewer #1: Yes

3. Has the statistical analysis been performed appropriately and rigorously? 

Reviewer #1: No

4. Have the authors made all data underlying the findings in their manuscript fully available?

Reviewer #1: Yes

5. Is the manuscript presented in an intelligible fashion and written in standard English?

Reviewer #1: Yes

6. Review Comments to the Author

Reviewer #1: (No Response)

7. PLOS authors have the option to publish the peer review history of their article (what does this mean?). If published, this will include your full peer review and any attached files.

Reviewer #1: No

---

## [Editor Report · Acceptance letter]

PONE-D-25-63192R1

PLOS One

Dear Dr. Luo,

I'm pleased to inform you that your manuscript has been deemed suitable for publication in PLOS One. Congratulations! Your manuscript is now being handed over to our production team.

Kind regards,

on behalf of

Dr. Ali Faramarzi

Academic Editor

PLOS One